# Region-Aware Generalized Face Anti-Spoofing via Chebyshev Convolutional Graph Networks

## Abstract

Face Anti-Spoofing (FAS) plays a crucial role in safeguarding face recognition systems from adversarial attacks. Current approaches leveraging Convolutional Neural Networks (CNNs) and Vision Transformers encounter challenges in generalizing across diverse attack behaviors and region-specific variations. These limitations stem from: (1) the heterogeneous characteristics of attacks across different facial regions, which arise due to varying color, texture, and material properties, and (2) the expansive data space, complicating effective generalization. To address these issues, we propose a novel approach using Chebyshev Convolutional Graph Neural Networks (ChebConv GNN), which excels in capturing spatial information within graph structures. Specifically, ChebConv efficiently processes graphs constructed from image data, allowing for precise modeling of visual features. We begin by processing regions around facial landmarks through the initial layers of DenseNet to extract node features, capturing localized and rich information from each region. Each facial region is assigned a node, forming a unified graph where corresponding nodes across faces represent the same regions. This design enables the network to adapt dynamically to region-specific features while modeling inter-regional relationships effectively, reducing the data space and improving generalization. To further enhance domain adaptation, we introduce a Domain-Adversarial Graph Network, which bolsters performance across unseen domains. Additionally, we incorporate a self-supervised auxiliary task to promote the learning of texture features in each region, strengthening the model's ability to differentiate between genuine and spoofed faces. Experimental results demonstrate that our approach not only improves accuracy but also significantly enhances generalization, surpassing the performance of existing methods. The code for the model and the results can be found at the following link: https://github.com/hassanyousefzade/RA-FAS.git.

## 1 Introduction

Face Anti-Spoofing (FAS) plays a crucial role in protecting facial recognition systems against various presentation attacks, such as printed photos, replayed videos, and 3D masks Kunert (2023); Greenberg (2017). Although existing methods for Presentation Attack Detection (PAD) George & Marcel (2019); Liu et al. (2022a; 2023a; 2018; 2020); Yu et al. (2020a); Zhang et al. (2020a) perform well in intra-dataset experiments, their performance significantly drops when confronted with unseen domainsChen et al. (2021); Wang et al. (2022a;e); Zhou et al. (2022; 2024). This issue arises due to the large distributional discrepancies between different domains, which increases the security challenges for facial recognition systems. Therefore, the development of robust FAS methods is essential to enhance the security of facial recognition systems.

Spoof detection initially relied on handcrafted features like SIFT Patel et al. (2016), LBP Boulkenafet et al. (2015); de Freitas Pereira et al. (2013), and HOG Komulainen et al. (2013); Yang et al. (2013b). With the advent of deep learning, researchers shifted their focus to deep neural networks for feature extraction in spoof detection Yang et al. (2014); Feng et al. (2016b); Zhang et al. (2021); Li et al. (2016). Despite these advancements, challenges related to performance in unseen domains and handling distribution shifts remain. To address these issues, Domain Generalization (DG) tech-

niques have been extensively introduced into Face Anti-Spoofing (FAS) tasks to mitigate the effects of domain discrepancies Liu et al. (2022c; 2024); Wang et al. (2019). Popular techniques include Domain Adversarial Learning Jia et al. (2020a); Kwak et al. (2023); Wang et al. (2022e); Shao et al. (2019c), Meta-Learning Chen et al. (2021); Du et al. (2022); Jia et al. (2021), Feature Disentangling Liu et al. (2022b); Zhang et al. (2020a), and Contrastive Learning Wang et al. (2022e). Although these methods aim to learn domain-invariant features, challenges such as poor performance in domains that significantly differ from the training data still persist. Despite recent successes in the use of Convolutional Neural Networks (CNNs) and more recently Vision Transformers (ViTs) in FAS George & Marcel (2021); Hong et al. (2023); Huang et al. (2023); Liao et al. (2023); Liu & Liang (2023); Liu et al. (2023b); Wang et al. (2022b;c), these methods have difficulty modeling the spatial information and texture variations across different facial regions. Due to the variations in color, texture, and physical properties across different facial regions, spoofing attacks may exhibit different behaviors in different parts of the face. For example, as shown in Figure 1, the eye re-

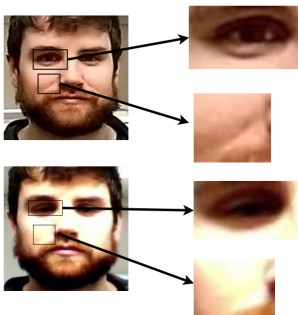

Figure 1: In this example, you can observe that the brightness of the cheeks in the attack face is higher compared to the original face, which is not the case in the original image. Additionally, the hair in the attack face appears darker. Some details in specific areas, such as the cheeks, have disappeared due to the attack, as shown in the figure. For instance, the eyes have become darker. As a result, each part of the face responds uniquely to the specific attack.

gions in the attack image appear darker compared to the eyes in the original face. Additionally, the wrinkles on the cheeks have disappeared in the attack image, and the brightness of the image has noticeably increased compared to other areas of the face. Furthermore, the hair in the attack face appears darker. To address this problem, we propose a method based on Chebyshev Graph Neural Networks (ChebConv GNNs) Defferrard et al. (2016). In this approach, graph nodes are assigned to specific regions of the face, allowing the model to adjust the behavior of each node according to the position and texture of the facial region. Each node has its own distinct pattern and is placed within a facial graph, helping the model better capture regional features. This not only reduces unnecessary diversity in the features but also significantly improves the generalization ability of the model. Furthermore, the facial graph we extract assigns each node to a specific region of the face. As a result, each node shares a similar pattern with the corresponding node in other faces, leading to a shared semantic subspace between the graphs. This shared subspace helps reduce unnecessary feature diversity and allows the proposed Graph-based Domain Adversarial Learning to optimally learn this shared subspace. Additionally, we define a self-supervised auxiliary task to extract facial texture features, which helps distinguish facial regions in the feature space and enhances the texture features containing spoofing and liveness information. In this work, we introduce several significant contributions to the field of face anti-spoofing: **First Use of Graph Neural Networks:** We explore the application of Chebyshev Graph Neural Networks for face anti-spoofing, offering a fresh perspective that enhances our understanding of facial features compared to traditional methods. **Focus on Localized Features:** Our approach emphasizes the analysis of specific facial regions, which is crucial since different areas exhibit unique traits that help the system better identify potential spoofing attempts. **Introduction of Node-level Auxiliary Tasks:** We propose a novel auxiliary task where each facial region identifies its specific area (such as the lips or cheeks). This encourages the model to learn about the interactions of different features, significantly boosting overall accuracy. **Stronger Generalization and Cross-domain Accuracy:** Our method demonstrates improved generalization capabilities, achieving higher accuracy in diverse testing scenarios compared to many existing techniques.

## 2 PROPOSED METHOD

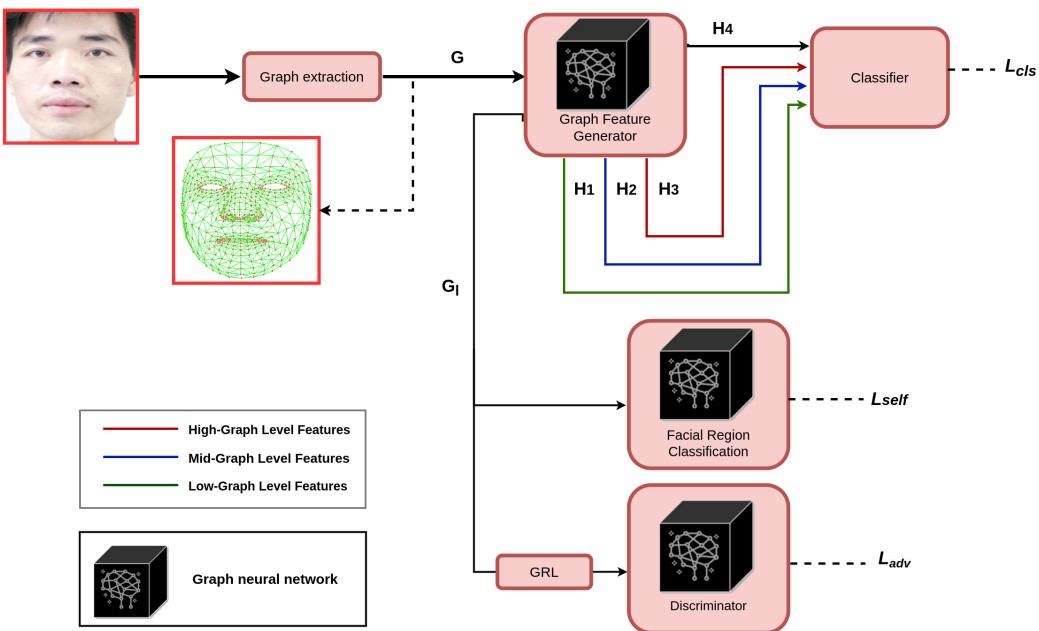

Figure 2: In the above figure, the image is first transformed into a graph(graph extraction module). Then, the extracted graph(G) is sent to a graph feature generator network. This block has several outputs. The first output($G_l$) is a graph, which is sent to the GRL block. Additionally, this graph-level output is fed into the facial region classification block.The H1, H2, H3 , H4 outputs are graph-level embeddings, which are sent to the classifier and used for the final classification. Finally, all three losses are combined as the final output.

In this section, we introduce our method, illustrated in Fig. 2. First, we provide a brief overview of the Chebyshev Graph Convolutional Network (ChebConv). Second, we describe the process of extracting a graph representation from facial images. Third, we input this graph into a graph graph feature generator network, which combines local features specific to each node and generates global features. Subsequently, global pooling is applied across multiple layers of the graph to extract both high and low graph level representations. Fourth, these representations are concatenated and passed to a classifier for final prediction. In the fifth step, the generated first-level graph is fed into an adversarial domain adaptation graph neural network to improve generalization to unseen domains. Additionally, a self-supervised auxiliary task is incorporated to further improve the learning of facial texture features. Finally, the overall loss is integrated to optimize the network, ensuring stable and reliable training.

### 2.1 CHEBYSHEV GRAPH CONVOLUTIONAL NEURAL NETWORKS

In this subsection, **Chebyshev Convolutional Neural Networks (ChebConv)** are introduced as an effective method for processing graph data, particularly for graphs extracted from imagesDefferrard et al. (2016). The main idea behind ChebConv is to extend traditional convolution operations to irregular graph structures, enabling improved modeling of both local and spatial information. These networks utilize local filters based on Chebyshev polynomials and graph Laplacians.

In our proposed method, each node corresponds to a specific region of the face, derived from **facial key points**, and a graph is constructed to connect these regions. By increasing the value of $K$, the filters can aggregate more information from neighboring nodes, which enhances the learning of both local and global features.

One of the main advantages of ChebConv over **traditional Graph Convolutional Networks (GCNs)** is the **reduction in computational complexity**. This reduction in complexity significantly

improves the efficiency and speed of the model, especially in the context of large and complex graphs.

This architecture is particularly effective for graphs extracted from images, allowing the model to effectively capture intricate spatial relationships between different facial regions.

## 2.2 GRAPH EXTRACTION FROM FACE

In this subsection, we describe the process of **graph extraction from images**, as illustrated in Figure 2a. To extract **facial key points** from the images, we utilized **Mediapipe** AI (2024), a tool capable of detecting facial landmarks with high speed and accuracy. Mediapipe extracts 468 key points from the face, which are treated as nodes in our graph structure.

Once these key points are identified, a region around each point is defined. The size of the window for each region is set to **40×40 pixels**. This results in 468 image patches $I_i$ for $i = 1, 2, ..., 468$. We utilize the following operations for feature extraction, referred to collectively as **D121-LowFE** (DenseNet-121 Low Feature Extractor):

- **DenseNet-121 (D)**: We use the **first two Dense blocks of DenseNet-121 Huang et al. (2017), pre-trained on ImageNet** (abbreviated as **D121-2B**) for feature extraction. The functions are represented as $f_{DenseBlock1}$ for the first Dense block and $f_{DenseBlock2}$ for the second Dense block.
- **Global Average Pooling (G)**: The outputs from these two Dense blocks are passed through a **Global Average Pooling (GAP)** layer to reduce the feature dimensions.
- **Flattening (F)**: The resulting features are then **flattened** to prepare them for the next processing stage.

The operations are defined as follows:

$$f_i^{(1)} = f_{DenseBlock1}(I_i) \tag{1}$$

$$f_i^{(2)} = f_{DenseBlock2}(I_i) \tag{2}$$

Initially, the outputs from the first and second Dense blocks are passed through a **Global Average Pooling (GAP)** layer:

$$G_i^{(1)} = GAP(f_i^{(1)}) \tag{3}$$

$$G_i^{(2)} = GAP(f_i^{(2)}) \tag{4}$$

The two outputs are then **concatenated**:

$$G_i = Concat(G_i^{(1)}, G_i^{(2)}) \tag{5}$$

Finally, the concatenated output $G_i$ is **flattened**:

$$F_i = Flatten(G_i) \tag{6}$$

These flattened feature vectors are used as the **node features** for the next stage of the model.

The **edges** in the graph are directly derived based on the connections extracted by **Mediapipe** and are binary. Specifically, $E_{ij}$ represents the presence or absence of a connection between nodes $i$ and $j$. If nodes $i$ and $j$ are connected, $E_{ij} = 1$, otherwise $E_{ij} = 0$:

$$E_{ij} = \begin{cases} 1 & \text{if nodes } i \text{ and } j \text{ are connected,} \\ 0 & \text{if nodes } i \text{ and } j \text{ are not connected.} \end{cases} \tag{7}$$

This structure ensures consistency with the facial structure, as shown in Figure 2c.

The selection of **D121-LowFE**'s early layers is motivated by their ability to effectively capture **texture features** and **details relevant to spoofing**. Furthermore, these features are more **generalized** and less dependent on the **ImageNet classification task**. Additionally, due to the **shallower nature** of these layers, they offer a **significant speed advantage**, making the proposed method more efficient.

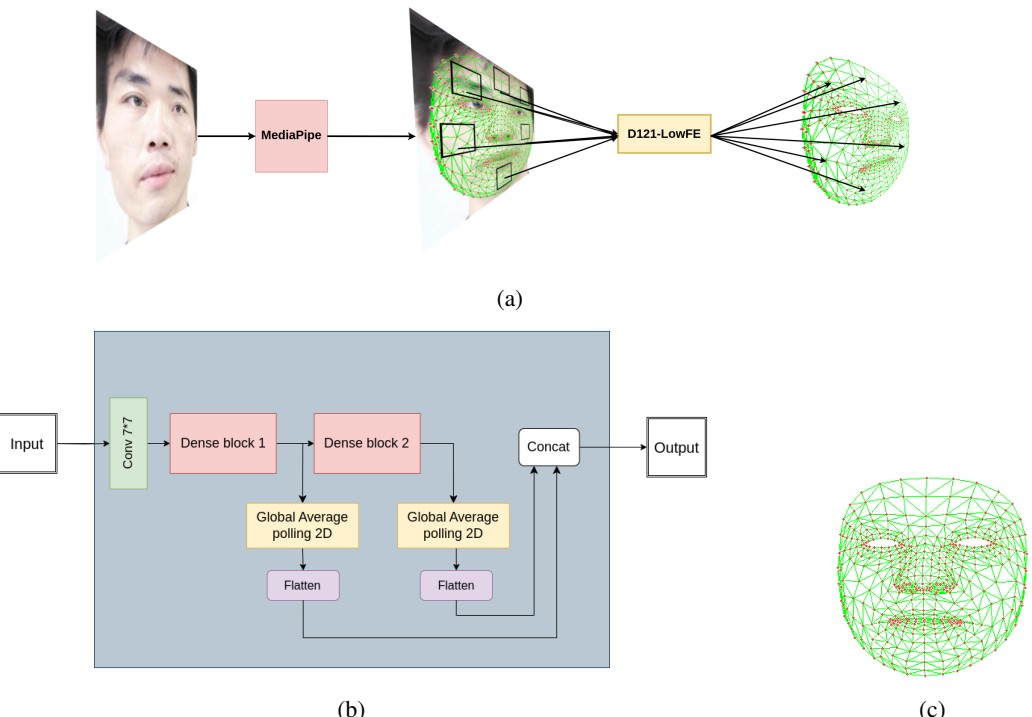

Figure 3: (a)Here, the input face image is first passed through MediaPop, from which a graph is extracted. Then, using D121-lowFE, the low-level features related to the textures of each facial region are extracted and considered as node features.(b)As shown in the figure above, we first select the initial two layers of DenseNet121. Then, we apply global average pooling to reduce their dimensions, flatten them, and finally concatenate them.(c)The graph that MediaPipe provides from the input face image.

### 2.3 GRAPH FEATURE GENERATOR

The **Graph Feature Generator** is designed to efficiently capture and process node features in a graph-based structure. As shown in Figure 3, this process is based on stacking **six Chebyshev Convolutional Blocks (Cheb Blocks)** illustrated in Figure 3a, where each block contains the following components:

The first component of each Cheb Block is the **Chebyshev Graph Convolution** layer, which aggregates information from neighboring nodes. The node features at layer $l + 1$ are updated using the following equation:

$$h^{(l+1)} = \sum_{k=0}^{K-1} \theta_k T_k(\tilde{L}) h^{(l)}$$

where $h^{(l)}$ is the node feature vector at layer $l$, $\tilde{L}$ is the normalized Laplacian matrix of the graph, $T_k(\tilde{L})$ are the Chebyshev polynomials of degree $k$, $\theta_k$ are the learnable weights corresponding to each degree $k$, $K$ is the filter size determining how many neighbors are considered in the graph, and The initial node features are represented as $h^{(0)} = F$.

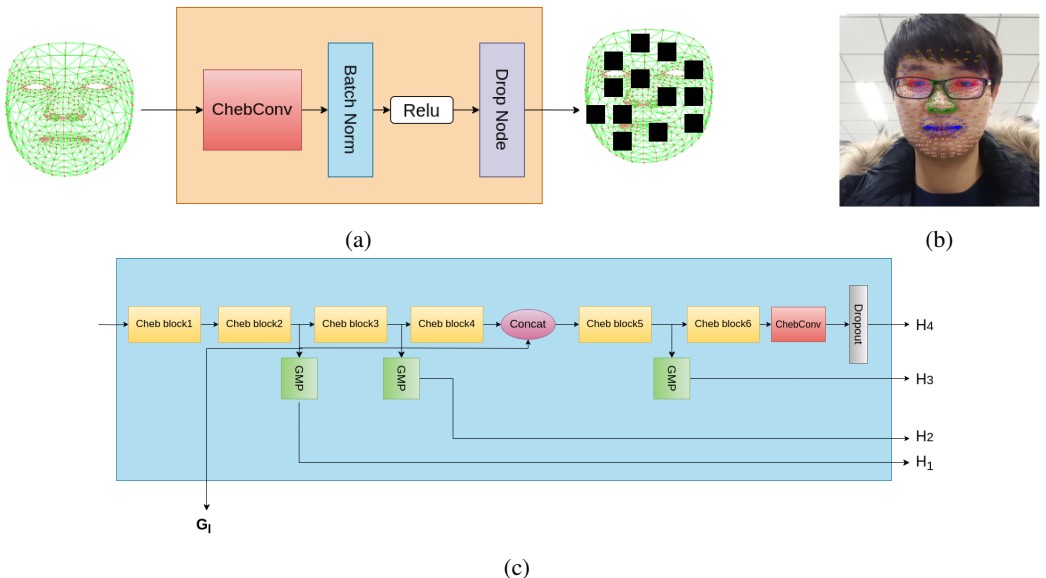

Figure 4: (a) In this figure, you can see the Cheb block, which takes a graph as input. First, it applies a Chebyshev Graph Convolutional Neural Network (GCN) layer, followed by a Batch Normalization layer, a ReLU activation function, and finally a Dropout layer. (b) In this figure, you can see the overall architecture of the feature-generating graph, which consists of 6 Cheb Block layers. This architecture has five outputs. The first input is a graph $G_l$, used as input for the GRL block. The second to fifth inputs are $H_1, H_2, H_3, H_4$, which are fed into the classifier. Additionally, to prevent the loss of texture and spatial information, this graph is also fed into the face region classifier network, which is responsible for enhancing these features. (c) In this figure, you can see the 7 regions that have been selected as self-supervised targets.

Once the convolution operation is completed, the output is normalized using **Batch Normalization** to stabilize the learning process:

$$h^{(l+1)} = \text{BatchNorm}(h^{(l+1)})$$

After normalization, the model introduces non-linearity using the **ReLU activation function**:

$$h^{(l+1)} = \text{ReLU}(h^{(l+1)})$$

In this model, the initial node features $h^{(0)}$ are denoted as $F$, which represents the initial feature vector of each node before any message passing occurs. This initial feature vector for each node is obtained in the previous step. The Chebyshev Graph Convolution layer enables efficient propagation of information across the graph, capturing both local and global node relationships.

To further enhance generalization and prevent overfitting, we apply a **DropNode** layer at the end of each Cheb Block. **DropNode** randomly zeroes out some of the node features with a probability $P$, defined as:

$$h_i^{(l+1)} = \begin{cases} 0 & \text{with probability } P \\ h_i^{(l)} & \text{with probability } 1 - P \end{cases}$$

where $h_i^{(l)}$ is the feature of node $i$ at layer $l$, and $P$ is the probability of dropping the node.

By randomly deactivating nodes, **DropNode** prevents the model from becoming too reliant on specific nodes and ensures better generalization to unseen data. Nodes that are zeroed out do not participate in message passing in the subsequent layers, which further helps in reducing overfitting.

To increase the depth of the network and enhance its capacity, we utilize **six Chebyshev Blocks (Cheb Blocks)** (Figure 3b). It is known that in the initial layers of the network, the node embeddings are more local, while as we move closer to the end of the network, these embeddings become more general and comprehensive.

To leverage these embeddings at the graph level, we apply **Global Average Pooling** on the node features to achieve graph-level embeddings. In other words, the embeddings obtained from the initial, intermediate, and final layers are considered as outputs, allowing the classifier to utilize the embeddings from all three levels (local, semi-global, and global) of the graph.

Additionally, we employ **Skip Connections** to mitigate the phenomenon of over smoothing, which is common in graph neural networks. This process enables the model to effectively capture both local and global node features in a multi-level manner, leading to improved generalization.

## 2.4 CLASSIFIER

The multi-level graph level embeddings obtained from the previous stage, denoted as $H_1$, $H_2$, $H_3$, and $H_4$, are concatenated to form the input feature matrix $H_{\text{input}}$. The concatenation of these features can be represented as:

$$H_{\text{input}} = [H_1, H_2, H_3, H_4] \tag{1}$$

As a result, the input feature matrix $H_{\text{input}}$ contains embeddings from four levels. The number of tokens input to the Multi-head Transformer, denoted by $T$, is equal to the graph node embedding size $d$:

$$T = d \tag{2}$$

The Multi-head Transformer is applied to the input feature matrix to enhance the model's capacity and improve its ability to separate and distinguish features Li et al. (2020):

$$Z = \text{Transformer}(H_{\text{input}}) \tag{3}$$

Here, $Z$ represents the output features after applying the multi-head attention mechanism.

The output of the transformer is passed through a sigmoid activation for binary classification (spoof vs. live). The binary cross-entropy loss for a mini-batch of $N$ samples is defined as:

$$L_{\text{BCE}} = -\frac{1}{N} \sum_{i=1}^{N} [y_i \log(\hat{y}_i) + (1 - y_i) \log(1 - \hat{y}_i)] \tag{4}$$

Where $y_i \in \{0, 1\}$ is the ground truth label (1 for live, 0 for spoof) for the $i$-th sample. $\hat{y}_i$ is the predicted probability for the $i$-th sample.

## 2.5 ADVERSARIAL LEARNING FOR DOMAIN-INVARIANT GRAPH REPRESENTATION

We assume that there are minor distributional differences across different domains based on the following observations:

1. Considering samples from various domains, all graphs contain nodes corresponding to specific facial regions, and each node maintains a similar pattern in the same region. Therefore, these graphs share a common semantic feature space.

2. Both real-world samples and spoofing attacks often exhibit similar physical characteristics, such as shape and size.

Based on this, we employ adversarial learning to make the generated graphs indistinguishable across different domains.

Specifically, the parameters of the **Feature Generator Graph** ($\mathcal{G}_G$), which only uses the first-level embeddings (due to their more localized node features), are optimized by maximizing the adversarial loss function, while the parameters of the **Domain Discriminator Graph** ($\mathcal{D}_G$) are optimized in the opposite direction. This process can be formulated as follows:

$$\min_{G_G} \max_{D_G} L_{adv}(G_G, D_G) = -\mathbb{E}_{(x,y)\sim(X,Y_D)} \sum_{i=1}^{M} \mathbf{1}[i = y] \log D_G(G_G(x)),$$

where $Y_D$ is the set of domain labels, $M$ is the number of different domains, and $G_G$ and $D_G$ denote the Feature Generator Graph and Domain Discriminator Graph, respectively. To optimize $G_G$ and $D_G$ simultaneously, we use a **Gradient Reversal Layer (GRL)** Ganin & Lempitsky (2015b), which inverts the gradient by multiplying it by a negative value during the backpropagation step. This allows the parameters of $G_G$ and $D_G$ to be optimized in opposing directions. For the **Domain Discriminator Graph**, we utilize **GraphSAGE** Hamilton et al. (2017), which takes the first-level embeddings from the Feature Generator Graph as input. After passing through several GraphSAGE layers, the output undergoes **Global Average Pooling (GAP)** to obtain graph-level embeddings. These embeddings are then fed into a fully connected (dense) layer to classify the domain.

## 2.6 SELF-SUPERVISED AUXILIARY TASK FOR TEXTURE ENHANCEMENT

To enhance texture-related features and better differentiate various facial regions for more accurate texture feature extraction, we define a self-supervised auxiliary task. This task receives the first-level graph from the **Feature Generator** as input, and after passing through two **GraphSAGE** layers, it is fed into a **Multi-Head Transformer**. The transformer then performs classification to segment the face into eight distinct regions, as shown in Figure 5. The learning process for this self-supervised task can be formulated as follows:

$$L_{self} = -\sum_{i=1}^{N} \sum_{c=1}^{C} y_i^c \log(p_i^c)$$

where $N$ is the number of nodes (or samples), $C$ is the number of classes (or facial regions) $y_i^c$ is the ground truth label for node $i$ in class $c$, and $p_i^c$ is the predicted probability for node $i$ and class $c$. This formula represents the cross-entropy loss function used for the classification task, where the model maximizes the probability of correctly predicting each node's class, ensuring accurate facial region classification.

## 2.7 TOTAL LOSS

Next, the sum of all losses is calculated. The weights for each loss are set to 1. The total loss is computed as follows:

$$L_{total} = L_{BCE} + L_{adv} + L_{self}$$

This total loss is propagated throughout the network, and optimization is performed accordingly.

## 3 EXPERIMENTAL SETUP

**Databases.** Based on previous works in Domain Generalization (DG) for Face Anti-Spoofing (FAS) Jia et al. (2020a); Liu et al. (2021a;c), we evaluate our proposed method and compare it with other approaches using four publicly available FAS databases: OULU-NPU Boulkenafet et al. (2017) (denoted as O), CASIA-FASD Zhang et al. (2012) (denoted as C), MSU-MFSD Wen et al. (2015) (denoted as M), and Idiap Replay-Attack Chingovska et al. (2012) (denoted as I). These datasets were collected under various conditions, including different devices, attack types, lighting conditions, and backgrounds, leading to significant domain shifts. Our experiments strictly follow the protocols used in previous DG-based methodsLiu et al. (2021a;c); Wang et al. (2022a;e). We utilize three metrics to evaluate the model's performance: 1. **HTER** (Half Total Error Rate): It computes the average of the False Rejection Rate (FRR) and False Acceptance Rate (FAR). 2. **AUC** (Area Under the ROC Curve): This metric assesses the theoretical performance of the model. 3. **TPR** at a fixed

Table 1: Comparison of HTER (%) and AUC (%) on four public FAS datasets under different domain generalization settings.

| Method | OCI→M | | OMI→C | |
|---|---|---|---|---|
| | HTER↓ | AUC | HTER↓ | AUC |
| MADDG Shao et al. (2019b) | 17.69 | 88.06 | 24.50 | 84.51 |
| DR-MD-Net Jia et al. (2020b) | 17.02 | 90.10 | 19.68 | 87.43 |
| RFMeta Shao et al. (2019b) | 13.89 | 93.98 | 20.27 | 88.16 |
| NAS-FAS Yu et al. (2020b) | 13.59 | 88.63 | 16.54 | 90.18 |
| D$^2$AM Chen et al. (2021) | 12.70 | 95.06 | 20.98 | 85.58 |
| SDA Wang et al. (2021) | 15.40 | 91.80 | 24.50 | 84.50 |
| DRDG Zhou et al. (2023a) | 12.43 | 95.81 | 19.05 | 88.79 |
| ANRL Liu et al. (2021b) | 10.83 | 96.15 | 14.79 | 89.12 |
| SSDG-R Jia et al. (2020a) | 7.38 | 97.17 | 10.44 | 95.94 |
| SSAN-R Wang et al. (2022d) | 6.67 | 98.75 | 10.00 | 96.67 |
| PatchNet Wang et al. (2022a) | 7.10 | 98.46 | 11.34 | 94.58 |
| SA-FAS Sun et al. (2023) | 5.95 | 96.55 | 8.78 | 97.58 |
| IADG Zhou et al. (2023a) | 5.41 | 98.19 | 8.70 | 96.44 |
| GAC-FAS Zhou et al. (2023c) | 5.0 | 97.56 | 8.20 | 95.16 |
| HPDR Zhou et al. (2023d) | 4.58 | 96.02 | 11.30 | 94.42 |
| TTDG-V Zhou et al. (2024) | 4.16 | 98.48 | 7.59 | 98.18 |
| RAFAS (Ours) | **3.84** | **98.97** | **5.11** | **98.62** |

False Positive Rate (FPR=1%): This metric is useful for selecting an appropriate threshold based on the specific requirements of real-world applications.

**Implementation Details** The embedding size of nodes in the graph, both for the feature generator graph and the discriminator, as well as for the facial region classifier, is set to 128. The value of K is also set to 2. For optimization, the Adam optimizer with a learning rate of $1e - 4$ is used. The batch size is set to 64, and the dropNode probability is set to 0.3. In the feature generator graph, the first and second blocks have a dropNode rate of zero, as applying dropNode in these blocks harms the performance of the facial region classifier. Additionally, the input images are cropped using MediaPipe and resized to 256x256. The experiments are conducted for a maximum of 500 epochs. The implementation of graph neural networks was done using the **PyTorch Geometric** Contributors (2024) library and the **PyTorch** Paszke et al. (2017) framework. The computations were carried out on an **RTX 3090** hardware.

## 3.1 COMPARISON OF RESULTS

Table 2: Comparison of HTER (%) and AUC (%) on two source domains and one target domain under specific domain generalization settings.

| Method | MI → C | | MI → O | |
|---|---|---|---|---|
| | HTER↓ | AUC↑ | HTER↓ | AUC↑ |
| MSLBP Määttä et al. (2011) | 51.16 | 52.09 | 43.63 | 58.07 |
| Color Texture Boulkenafet et al. (2015) | 55.17 | 46.89 | 53.31 | 45.16 |
| LBPTOP Pereira et al. (2013) | 45.27 | 54.88 | 47.26 | 50.21 |
| MADDG Shao et al. (2019b) | 41.02 | 64.33 | 39.35 | 65.10 |
| SSDG-M Jia et al. (2020a) | 31.89 | 71.29 | 36.01 | 66.88 |
| D$^2$AM Chen et al. (2021) | 32.65 | 72.04 | 27.70 | 75.36 |
| DRDG Liu et al. (2021c) | 31.28 | 71.50 | 33.35 | 69.14 |
| ANRL Liu et al. (2021a) | 31.06 | 72.12 | 30.73 | 74.10 |
| SSAN Wang et al. (2022d) | 30.00 | 76.20 | 29.44 | 76.62 |
| EBDG Du et al. (2022) | 27.97 | 75.84 | 25.94 | 78.28 |
| AMEL Liu et al. (2021d) | 24.52 | 82.12 | 19.68 | 87.01 |
| IADG Zhou et al. (2023a) | 24.07 | 85.13 | 18.47 | 90.49 |
| GAC-FAS Le & Woo (2024) | 16.91 | 88.12 | 17.88 | 89.67 |
| RAGFAS(Ours) | **11.23** | **91.52** | **13.46** | **90.14** |

Table 3: The impact of different loss functions on the model's performance in the MO to C setting. The table shows that adding the $L_{Adv}$ loss function significantly improves performance compared to adding $L_{Self}$ or the Drop Node layer.

| $\mathcal{L}_{Adv}$ | $\mathcal{L}_{Self}$ | DropNode | HTER(%) | AUC(%) |
|---|---|---|---|---|
| | | | 21.60 | 87.58 |
| ✓ | | | 14.47 | 92.14 |
| ✓ | ✓ | | 9.15 | 96.32 |
| ✓ | ✓ | ✓ | **8.87** | **96.60** |

In this work, following prior research Chengyang Hu & Ma (2022); Jia et al. (2020a); Liu et al. (2021a;c), we evaluate various methods using three evaluation settings: **Leave-One-Out (LOO)** validation, **Limited Source Domains**, and **Cross-Attack** scenarios to assess the models' generalization ability to unseen domains and attacks.

**Leave-One-Out (LOO) Validation:** In the **Leave-One-Out (LOO)** setting, three datasets are utilized as source domains for training, leaving the remaining dataset as the target domain for testing in the face anti-spoofing (FAS) task.

As shown in Table 1, our method achieves state-of-the-art performance in domain generalization, outperforming previous approaches. Specifically, *Graph-based Domain Adversarial Learning* successfully captures a shared feature space, while *Graph-based Self-Supervised Learning* effectively extracts domain-independent texture features. To further highlight the effectiveness of these techniques, we compare model performance with and without their use. Results in Table 1 demonstrate that our model constructs a graph that maps data into a shared semantic space, reducing irrelevant feature variance and significantly enhancing generalization.

**Limited Source Domains:** To evaluate the robustness of our method under constrained conditions, we assess its performance when trained with a limited number of source domains. Based on previous studies Liu et al. (2021a;c), the **MSU** and **Idiap** datasets are used as source domains, while the **OULU** and **CASIA** datasets are employed for testing. As shown in Table 2, our method demonstrates significantly better performance in limited domains compared to other approaches, showcasing superior generalization in these scenarios.

### 3.2 ABLATION STUDY

**Different Loss Functions.** We conducted all our experiments based on the MO to C settings. We examined our loss functions hierarchically; first, we added the loss function related to $L_{adv}$, and then we added the loss function related to $L_{self}$. After that, we also experimented by adding a Drop Node layer. As shown in Table 3 in the appendix, adding $L_{adv}$ had a significantly greater impact compared to the other two.

### 3.3 VISUALIZATION AND ANALYSIS

Moreover, to demonstrate that our method works logically, we used the Grad-CAM technique in graph neural networks to show which nodes the network pays more attention to. As shown in Figure 5, our method focuses on nodes where spoofing is clear and evident.

## 4 CONCLUSION

The RAGFAS method is based on modeling the behavioral differences of attacks across different parts of the face and utilizes graphs and Chebyshev Convolutional Graph Neural Networks. By employing a self-supervised technique, we prevented the removal of texture and spatial features related to the face during the learning process, which significantly improved the accuracy of the model. Our experiments demonstrate that this approach not only performs well in the three source domains but also outperforms other methods by a significant margin in limited domains.

It has been able to reduce unnecessary feature variations in the data while significantly improving generalization performance.

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

## A  APPENDIX

## B  RELATED WORK

In the early stages of research, handcrafted features were widely employed for detecting face spoofing attacks. These features include LBP Pereira et al. (2014), HOG Feng et al. (2016a), and SIFT Yang et al. (2013a). Concurrently, studies investigated predefined biometric traits and behaviors such as blinking Mostaani et al. (2020), lip movement Zhou et al. (2023b), head rotation, and changes in facial expressions Ganin & Lempitsky (2015a). With the emergence of deep neural networks (DNNs), the detection capabilities of face anti-spoofing systems significantly improved Zhang et al. (2020b); Lin et al. (2019); Yu et al. (2021). These improvements were further facilitated by incorporating diverse types of supervisory inputs such as depth maps Zhang et al. (2020b), reflection maps Lin et al. (2019), and R-PPG signals Yu et al. (2021). Despite their success in intra-dataset scenarios, the performance of these methods tends to degrade in unseen domains. To tackle this, domain-generalization-based approachesShao et al. (2019a); Liu et al. (2021e) have been proposed to learn domain-invariant features through adversarial training or meta-learning. These approaches aim to align feature spaces across multiple domains. However, such direct alignment can neglect crucial discriminative information, especially when there is a large gap between domains. Additionally, self-supervised learning Liu et al. (2022d; 2023c) has been explored to reduce reliance on labeled data. Most of these techniques rely on convolutional neural networks (CNNs) for feature extraction. Recently, transformer-based models George & Marcel (2021); Hong et al. (2023); Huang et al. (2023) have shown remarkable success in face anti-spoofing. However, one of the major limitations of these approaches is their inability to fully capture the spatial relationships and texture variations across different regions of the face. To address this limitation, we propose a method based on Chebyshev Graph Neural Networks (ChebConv GNNs)Defferrard et al. (2016). In our approach, graph nodes correspond to specific facial regions, allowing each node to adapt its behavior according to the position and texture of its respective region. This design not only provides each node with a distinct feature pattern but also reduces unnecessary diversity (Diversity) in the features, thereby significantly improving the generalization (Generalization) of the model.

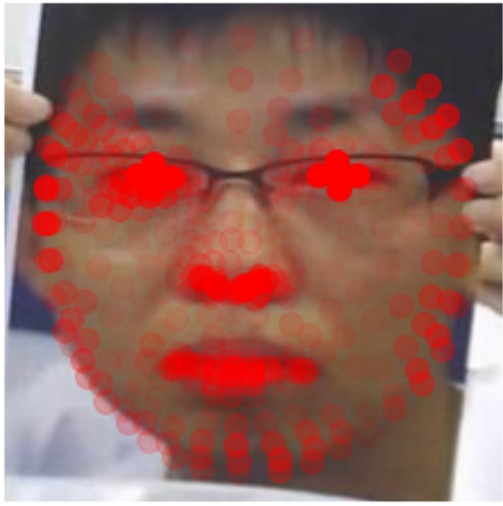

Figure 5: The Grad-CAM method in graph neural networks, where the darker the node, the higher its importance.

