# OpenReview forum: "Region-Aware Generalized Face Anti-Spoofing via Chebyshev Convolutional Graph Networks"
_ICLR.cc/2025/Conference — ICLR 2025 Conference Withdrawn Submission_

### Official Review · Reviewer_9nrt · 2024-10-28

**Soundness:** 2
**Presentation:** 2
**Contribution:** 2
**Rating:** 1
**Confidence:** 1

**Summary:**

This work discusses the challenges of Face Anti-Spoofing (FAS) in face recognition systems, especially in generalizing across diverse attacks and variations in facial regions. Traditional methods using CNNs and Vision Transformers struggle with these issues due to heterogeneous attack characteristics and the large data space. To address this, the authors propose using Chebyshev Convolutional Graph Neural Networks (ChebConv GNN), which efficiently capture spatial information in graph structures. Their approach models facial regions as nodes in a graph, allowing for dynamic adaptation to region-specific features and better generalization. Additionally, they introduce a Domain-Adversarial Graph Network for improved domain adaptation and a self-supervised auxiliary task to enhance texture feature learning. The proposed method demonstrates superior accuracy and generalization compared to existing techniques.

**Strengths:**

This work provides a new standpoint view that Chebyshev Convolutional Graph Neural Networks can provide more effective features for face anti-spoofing task.

**Weaknesses:**

I'm sorry, but it seems that the GitHub URL in this work—https://github.com/hassanyousefzade/RA-FAS.git—shows the author's ID and may conflict with the Anonymous URL requirements. Due to this concern, I prefer to reject this work.

**Questions:**

See weakness.

---

### Official Review · Reviewer_GP8K · 2024-11-04

**Soundness:** 2
**Presentation:** 1
**Contribution:** 1
**Rating:** 1
**Confidence:** 4

**Summary:**

This paper introduces a new approach to face anti-spoofing (FAS) using Chebyshev Convolutional Graph Neural Networks (ChebConv GNN) to overcome challenges in handling region-specific variations and generalizing to unseen domains. By assigning nodes to distinct facial regions, the model captures localized features through a graph structure, employs domain-adversarial learning and self-supervised auxiliary tasks to enhance cross-domain performance, and improves the model’s ability to differentiate between genuine and spoofed faces.

**Strengths:**

This paper introduces ChebConv GNN to the FAS task, creatively addressing existing limitations by focusing on the spatial relationships among distinct facial regions.
The paper effectively integrates domain adversarial learning and a self-supervised auxiliary task to enhance the model's performance and adaptability across varying domains.

**Weaknesses:**

There are several significant weaknesses that need to be addressed:

First, this paper has serious issues in basic writing, with numerous errors. For instance, the figures mentioned do not match the actual images: Fig 2 is referred to in the text but likely corresponds to Figure 2; Figure 2a may correspond to Figure 3; and Figure 3 and its subfigure Figure 3a may correspond to Figure 4. Additionally, the modules depicted in the figures lack detailed explanations in the text. For example, GRL (Graph Representation Learning) mentioned in Figure 1 is not specifically introduced in the corresponding section, leaving readers confused about its meaning. Furthermore, MediaPop mentioned in the title of Figure 3 does not appear in the paper and may refer to MediaPipe as intended by the authors. Regarding the title of Figure 4, “Figure 4: (a) In this figure, you can see the Cheb block, which takes a graph as input. First, it applies a Chebyshev Graph Convolutional Neural Network (GCN) layer,” it should be pointed out that the abbreviation of Chebyshev Graph Convolutional Neural Network as GCN is not adequately explained, which may cause confusion for readers.

Second, the paper lacks strong theoretical support in the introduction of its core content. The authors propose the use of Chebyshev Graph Neural Networks (ChebConv GNNs) to address the significant decline in discrimination performance on unseen domains in the field of face anti-spoofing, but they fail to reasonably justify why this method is effective. The paper only provides a general process for neural network data handling without necessary theoretical backing or empirical support, making it hard to be convincing. It is recommended that the authors strengthen the justification for the effectiveness of their method in future revisions to enhance the academic value of the paper.

**Questions:**

To better understand the contributions and methodology of the paper, I have the following questions:

(1)Clarification of RAGFAS: What is RAGFAS mentioned in the conclusion? It appears once in the experimental results in Table 2, but there is no detailed introduction explaining what it represents or what it stands for.

(2)Clarification of Figures: Could the authors clarify the discrepancies between the figures mentioned in the text and the actual figures presented? Specifically, please address the following:
     1.The references to Fig 2, Figure 2a, Figure 3, Figure 3a, and Figure 4, and their corresponding images.
     2.The meaning of GRL in Figure 1 and its relevance to the overall methodology.
     3.The term "MediaPop" in the caption of Figure 3, and whether it should be corrected to "MediaPipe."
     4.The rationale behind using the abbreviation GCN for Chebyshev Graph Convolutional Network in Figure 4.

(3)Clarification of Methodology: In line 140, the phrase “Third, we input this graph into a graph graph feature generator network” contains a repetition of the word "graph." Could the authors clarify this?

(4)Definition of K: In line 157, what does K represent when it states, “By increasing the value of K”?

(5)Computational Complexity: In line 160, could the authors explain why ChebConv can reduce computational complexity?

(6)Placement of Related Work: Why is the section on RELATED WORK placed in the APPENDIX?

(7)Theoretical Justification: The paper lacks a strong theoretical foundation for the proposed use of Chebyshev Graph Neural Networks (ChebConv GNNs). Can the authors provide a more detailed explanation of why this approach is particularly suited to address the challenges of performance decline in unseen domains for face anti-spoofing?

---

### Official Review · Reviewer_LoKt · 2024-11-04

**Soundness:** 2
**Presentation:** 2
**Contribution:** 2
**Rating:** 3
**Confidence:** 5

**Summary:**

This paper proposed a Chebyshev Convolutional Neural Networks for face  anti-spoofing. Experiments  conducted on the CIOM benchmarks  are presented. However, there  are nonnegligible writing issues, and this paper lack of indepth insight analysis of using  GCN for FAS. Thus, this paper did not reach the  bar  of ICLR, in my opinions.

**Strengths:**

The  performance looks good.

**Weaknesses:**

The application of  graph neural network is direct and there is missing novel  design with FAS-task insights. It is unclear for why some modules are used. For example, DenseNet module and Chebyshev Blocks, adversarial learning are available.  Simply combining these modules without  justifications and  novel insights  are not regarded as contribution

The citation format  is  incorrect: Please double check through the paper. For example, "We use the first two Dense blocks of DenseNet-121 Huang et al. (2017)" -> We use the first two Dense blocks of DenseNet-121 (Huang et al. 2017).

There are many writing issues, making this paper unreadable. For  example, Figure 5 is in the end  of  the paper, after the index.

**Questions:**

The  motivation  of the used modules  is unclear. Why is DenseNet used? Why is the transformerd used in the classifier?

Why Chebyshev GCN is useful for FAS?

---

### Official Review · Reviewer_L2Vd · 2024-11-04

**Soundness:** 2
**Presentation:** 2
**Contribution:** 2
**Rating:** 5
**Confidence:** 4

**Summary:**

To address FAS problem, this paper proposes to use Chebyshev Convolutional Graph Neural Networks (ChebConv GNN), which is expected to better capture spatial information within graph structures together with local features around pre-located facial landmarks. Experiments are conducted to show the effectiveness of the proposed method.

**Strengths:**

I recognize the following advantages of this submission:
1) to my knowledge, it is somewhat novel using graph neural network for FAS, which seems reasonable in capturing both the local features and the global structure of the facial parts. However, how much it depends on the accuracy of the facial landmarks is not explored.
2) from the experimental comparisons, the proposed method achieves better results, in case of domain generalization settings. This is important for the practical application of FAS techniques.

**Weaknesses:**

The proposed method relies on an off-the-shelf facial landmark localization module (Mediapipe AI) to obtain the local features, which in turn affect the node features. But, it is explored how much the accuracy of the facial landmarks will affect the proposed method. Experiments should be conducted.

For me, the datasets for evaluation of the techniques, as well as the evaluation protocols, in the field are hardly useful to support the advancement of the field. All the datasets were constructed before 2017, and the dataset sizes are relatively small. Of course, this is not the weakness of this submission. However, it is not difficult to involve more real faces for learning and testing with much lower false alarm rate.

The writing of the paper could be improved greatly. Sometime terms or concepts are used without definition. For example, on page 3, “By increasing the value of K...”, but K is not defined therebefore. In addition, some equations are too simple to have in a paper, for example, Equ. 1~6, which provides no useful information. The name of the proposed method is not consistent in different places of the paper, e.g. in Table 1 and Table 2.

**Questions:**

1. I would like to see experiments on how much the accuracy of the facial landmarks will affect the proposed method.
2. I would like to see evaluations with more real face images for testing under much lower FAR.

---

### Official Review · Reviewer_Ud1H · 2024-11-04

**Soundness:** 2
**Presentation:** 2
**Contribution:** 2
**Rating:** 3
**Confidence:** 5

**Summary:**

This paper proposes a novel Region-Aware Generalized Face Anti-Spoofing method using Chebyshev Convolutional Graph Neural Networks. The approach addresses challenges in generalizing across diverse spoofing attacks by capturing region-specific variations on facial features. The model enhances feature extraction through graph structures representing distinct facial regions, each of which is processed by dense convolutional layers for efficient feature generation. A domain adversarial component is incorporated to improve generalization across unseen domains, alongside a self-supervised auxiliary task for texture feature learning. Experimental results show that the proposed method achieves superior performance compared to existing techniques on several face anti-spoofing datasets.

**Strengths:**

The exploration of using GNN for face anti-spoofing

**Weaknesses:**

The GNN design appears somewhat rigid, with its core advantages insufficiently highlighted in the motivation and methodology, leaving its effectiveness to be further validated. Experimental verification is notably lacking. The paper writing is confusing.

**Questions:**

1. Using GNN for face anti-spoofing is not the first. [1] leverages the relationships between facial regions by GNN.
   *[1] Xu J, Lin W, Fan W, et al. A Graph Neural Network Model for Live Face Anti-Spoofing Detection Camera Systems[J]. IEEE Internet of Things Journal, 2024.*

2. The experimental protocol based on domain generalization is missing. Compared with existing face anti-spoofing methods, the experimental data presented in this paper does not show any improvement.
   *[2] Hu X, Liu H, Yuan H, et al. Fine-Grained Prompt Learning for Face Anti-Spoofing[C]//ACM Multimedia 2024. 2024.*

3. The benefit of local features for face anti-spoofing performance has already been demonstrated in many existing studies.

4. The statement in the second paragraph on page 2, "each node shares a similar pattern with the corresponding node in other faces," raises concerns. How can a node maintain a similar pattern if there are variations in data distribution or attack types?

5. The ablation studies are minimal. For instance, the reason why GNN can enhance face anti-spoofing performance has not been verified.

---

### Note · Authors · 2024-11-14

I have read and agree with the venue's withdrawal policy on behalf of myself and my co-authors.